# Exercise-Associated Hyponatremia in Endurance and Ultra-Endurance Performance–Aspects of Sex, Race Location, Ambient Temperature, Sports Discipline, and Length of Performance: A Narrative Review

**DOI:** 10.3390/medicina55090537

**Published:** 2019-08-26

**Authors:** Beat Knechtle, Daniela Chlíbková, Sousana Papadopoulou, Maria Mantzorou, Thomas Rosemann, Pantelis T. Nikolaidis

**Affiliations:** 1Medbase St. Gallen Am Vadianplatz, 9001 St. Gallen, Switzerland; 2Institute of Primary Care, University of Zurich, 8091 Zurich, Switzerland; 3Centre of Sports Activities, Brno University of Technology, 61669 Brno, Czech Republic; 4Department of Nutritional Sciences and Dietetics, International Hellenic University, 57001 Thessaloniki, Greece; 5Food Science and Nutrition Department, University of the Aegean, 81400 Myrina, Greece; 6Exercise Physiology Laboratory, 18450 Nikaia, Greece; 7School of Health and Caring Sciences, University of West Attica, 12243 Athens, Greece

**Keywords:** cold, heat, cerebral edema, prolonged exercise, swimming, cycling, running

## Abstract

Exercise-associated hyponatremia (EAH) is defined as a plasma sodium concentration of <135 mmol/L during or after endurance and ultra-endurance performance and was first described by Timothy Noakes when observed in ultra-marathoners competing in the Comrades Marathon in South Africa in the mid-1980s. It is well-established that a decrease in plasma sodium concentration <135 mmol/L occurs with excessive fluid intake. Clinically, a mild hyponatremia will lead to no or very unspecific symptoms. A pronounced hyponatremia (<120 mmol/L) will lead to central nervous symptoms due to cerebral edema, and respiratory failure can lead to death when plasma sodium concentration reaches values of <110–115 mmol/L. The objective of this narrative review is to present new findings about the aspects of sex, race location, sports discipline, and length of performance. The prevalence of EAH depends on the duration of an endurance performance (i.e., low in marathon running, high to very high in ultra-marathon running), the sports discipline (i.e., rather rare in cycling, more frequent in running and triathlon, and very frequent in swimming), sex (i.e., increased in women with several reported deaths), the ambient temperature (i.e., very high in hot temperatures) and the country where competition takes place (i.e., very common in the USA, very little in Europe, practically never in Africa, Asia, and Oceania). A possible explanation for the increased prevalence of EAH in women could be the so-called Varon–Ayus syndrome with severe hyponatremia, lung and cerebral edema, which was first observed in marathon runners. Regarding the race location, races in Europe seemed to be held under rather moderate conditions whereas races held in the USA were often performed under thermally stressing conditions (i.e., greater heat or greater cold).

## 1. Introduction

For years, exercise-associated hyponatremia (EAH) has been well-known among endurance and ultra-endurance athletes. Unfortunately, the occurrence of EAH continues to spread into a wider variety of sports and cause deaths in otherwise healthy individuals [1]. Athletes continue to die from complications associated with hyponatremic encephalopathy. In 2007, a 22-year-old male fitness instructor finished the London Marathon, where upon arrival, he collapsed and died due to EAH [2]. In 2015, a 30-year-old age group triathlete died after the Ironman Frankfurt due to EAH with hyponatremic encephalopathy [3].

The most likely reason for developing EAH is fluid overload [4,5] with the possibility of developing lung or brain edema [5,6] with fatal outcome [7]. Considering the increased number of endurance and ultra-endurance races, and of finishers in these races during the last years, comprehensive knowledge about EAH would be of great practical importance for athletes and professionals (e.g., coaches, nutritionists, practitioners, exercise physiologists) working with them. It should be highlighted that endurance and ultra-endurance athletes might compete in races differing for distance (e.g., marathon *versus* ultra-marathon) and under a wide range of environmental conditions (e.g., hot *versus* cold). Especially, it would be of interest to examine the variation of EAH by parameters such as sex, sport discipline, race distance and environmental conditions. 

The purpose of this review was to present new aspects regarding sex, race location, discipline and length of performance as risk factors in developing EAH. EAH has been described in both prolonged and non-prolonged exercise activities (e.g., yoga classes), yet it is more likely to develop in prolonged exercise [1,8]. A further aspect was to present data on the prevalence and severity of EAH regarding the discipline, environmental conditions, and length of the performance.

## 2. First Description and Definition of Exercise-Associated Hyponatremia

EAH describes the occurrence of hyponatremia in individuals during prolonged exercise (i.e., usually longer than six hours) and is defined when plasma or serum concentration of sodium is <135 mmol/L [9]. Apart from the definition of hyponatraemia based on biochemical severity [10], it can be also diagnosed based on symptomatology, as symptoms have been reported even at concentrations close to 130 mmol/L [11]. This can occur during or after prolonged physical exertion for 4–6 h or longer [12] and can be detected up to 24 h after the end of the exercise [13,14]. In the mid-1980s, Tim Noakes was the first to report severe symptomatic hyponatremia at the Comrades Marathon held in Durban, South Africa [15]. The Comrades Marathon is a 90-km long road-running race that has been held since 1921 between the South African cities of Durban and Pietermaritzburg. This race is the most traditional and participant-strong ultramarathon worldwide [16]. Hyponatremia was detected in four runners in 1981 and 1985 [17]. Prior to 1981, endurance athletes were advised not to drink during exercise [17,18], which in some cases led to hypernatremia [19]. Due to this knowledge, the American College of Sports Medicine (ACSM) recommended to drink as much as possible during exercise to prevent hypernatremia [17,18,20,21,22]. This advice led to an increased number of cases of EAH, especially in the USA. One major reason for this development was the US beverage industry and their funding of sports research with the aim of drinking as much as possible in order to avoid dehydration, which was measured by a loss in body mass during exercise [23].

## 3. The Pathogenesis of Exercise-Associated Hyponatremia

This is a condition as a common complication of especially endurance exercise due to a combination of over drinking beyond thirst and non-osmotic arginine vasopressin release [1]. Two factors determine the nature and the severity of the symptoms of hyponatremia: the speed of development and the level of plasma sodium. Rapidly developing hyponatremia tends to lead to central nervous symptoms (e.g., seizures) in the context of encephalopathy [24,25], as the cerebral adaptation mechanisms take effect only after a certain delay and therefore brain edema can develop [1]. It is important to know that the sodium concentration in the plasma rises only after prolonged exposure [26,27,28]. For example, Ironman triathletes who lost body mass had an increased sodium concentration, with the highest sodium concentration in athletes who had the highest body mass loss [29]. Two different mechanisms can lead to EAH [30]: on the one hand, increased sodium loss and on the other hand, increased water intake [31]. Athletes with symptomatic hyponatremia have an increased extracellular volume due to water retention [32,33].

Hyponatremia results from a dilution of a normal or slightly reduced total extracellular sodium concentration due to a loss of sodium via sweat and urine [34]. During exercise, there may be a large loss of sodium and fluid through urine and sweat, rarely vomiting [35]. When fluid is replenished with a sodium-free solution or hypotonic fluid, such as water, so that the extracellular fluid volume either remains the same or increases, hyponatremia will result from dilution [36,37,38]. The syndrome of inappropriate antidiuretic hormone secretion (SIADH) has been identified as a common cause of hyponatremia in clinical practice, though the diagnosis of SIADH and EAH should be differentiated with each other [39]. The failure of suppressing ADH has been identified in cyclists and marathon runners, and leads to water retention and hyponatremia, in conjunction with excessive fluid intake [1,40,41].

The body’s normal response to an uncorrected loss of sodium during exercise is a decrease in extracellular volume as a function of the sodium deficit [42,43], with long-distance athletes expected to develop normonatremia or even hypernatremia [44]. EAH is generally due to a relative fluid overload of the extracellular space [45]. EAH due to excessive sweat loss is likely very rare [46]. People have differing levels of sodium in their sweat; therefore, EAH due to high sodium loss could not be detected [35]. A case of a triathlon athlete and a cystic fibrosis patient have been described who had developed hypovolaemic EAH due to increased sweat loss and overconsumption of fluids relative to immediately available exchangeable sodium [46].

A case of a patient with cystic fibrosis has been described who has developed dilutional hyponatremia due to increased sweat loss during endurance exercise [46]. Three different mechanisms can lead to EAH, according to Noakes and colleagues [31]: excessive fluid drinking during exercise, retention of excess fluid because of inadequate suppression of antidiuretic hormone secretion, and osmotic inactivation of circulating sodium or failure to mobilize osmotically inactive sodium from internal stores. The current state of knowledge is that fluid overload with consecutive hyponatremia is the causative mechanism for EAH [31,32,47,48,49]. This dilutional hyponatremia is due to an increase in total body water relative to the total amount of exchangeable sodium in the body [12]. Although this increase may only be relative, in most cases of EAH there is an increase in body mass due to an increase in total body water [12,31].

On the basis of existing knowledge, it must be assumed that hyponatremia in the case of endurance exercise is due to excessive intake of predominantly sodium-poor or sodium-free liquids such as water [50]. This is the case when more fluid is consumed than sweated [51,52,53]. The large fluid intake leads to fluid retention in the body with dilutional hyponatremia [50,54]. It is important to mention also the existence of hypovolemic EAH, which would be predicted to develop in athletes exercising for more than 20 h and/or in hotter environment [55,56,57] and/or with higher sweat sodium losses [58]. There is a clear correlation between sodium concentration after a race and hydration during the race [59]. For example, EAH has been shown to occur when 3 L of water are drunk within 2 h of performance [60] or when 3 L are consumed overall, or when some fluid is consumed every mile [61]. It can happen that a triathlete consumes 16 L of fluid during an Ironman, gaining 2.5 kg of body mass and developing EAH [32,52].

Generally, fluid overload leads to an increase in body mass during endurance performance [31,62] and a decrease in plasma sodium [63]. The change in body weight correlates linearly to the plasma sodium concentration after exercise [31,64], and the relationship between the change in plasma concentration of sodium before and after exercise is inversely related to the change in body mass [65]. Often, athletes gain body mass and develop EAH [31,66]. Conversely, in long-distance competitions, it can also be seen that athletes lose body mass and have a high plasma sodium concentration [52]. The exception is hypovolemic EAH with a reduced body mass [56,57,67] indicating volume depletion. Consumption of electrolyte-free water during endurance exercise in the heat leads to a decrease of plasma sodium concentration [38]. It has been shown that body mass often remains unchanged and there is no fluid loss, but rather an excess in athletes with EAH [15,36]. The water remains in the intestine tract and does not move to the plasma [36]. Prolonged endurance performance with prolonged fluid intake leads to chronic intra- and extracellular hyperhydration with an increase in plasma volume [68]. However, many parameters remain the same (e.g., change in plasma sodium concentration, plasma osmolality, or body mass) for both mechanisms (i.e., sodium loss through sweating versus dilution due to fluid overload) [45]. In addition to fluid overload, inadequate suppression of secretion of the antidiuretic hormone (i.e., ADH, vasopressin) by a non-osmotic stimulation must also be considered for the development of EAH [1,31,48]. The increase in N-terminal pro-brain natriuretic peptide (NT-proBNP) shows that fluid overload is the cause of EAH [69].

SIADH is one of the mechanisms that lead to EAH [31]. Different stimuli have been recognised. In fact, an association between Interleukin-6 and arginine vasopressin has been observed, considering interleukin-6 to be the principal stimulator of arginine vasopressin in marathon runners with hyponatremia [55]. The authors highlight that SIADH is a pre-existing condition, in contrast to EAH due to fluid overconsumption, while also noting that NSAIDs play a role on anti-diuresis, and should be avoided before the race. In addition, it has been noted that-similarly to SIADH-athletes with EAH have submaximal suppression of ADH and high urine osmolality [9]. The suboptimal ADH according to the authors could be due to intense exercise, nausea or vomiting, hypoglycemia, pain, and emotion, as well as heat [1,9]. Furthermore, rhabdomyolysis associated acute renal failure could explain EAH [70,71], as does the increase in N-terminal pro-brain natriuretic peptide (NT-proBNP) [69]. Furthermore, Noakes and colleagues [31] also list the inactivation of osmotic sodium as one of the mechanisms of EAH development. The osmotically inactive exchangeable sodium stores were first identified by Edelman and colleagues [72]. These stores seem to be located in the bone, dense connective tissue, or cartilage [73]. In fact, bone loss has been associated with chronic hyponatremia in patients with SIADH [74]. The authors described sodium signalling mechanisms in osteoclasts, in order to mobilize sodium from bone stores during chronic hyponatremia, resulting in resorptive osteoporosis. Some athletes seem to mobilize sodium from internal osmotically inactive exchangeable sodium stores, while others may not be able to prevent the inactivation [31,75]. The improper inactivation of osmotically-active sodium might influence EAH development [76]. In fact, this has been speculated to be the case in athletes who developed EAH, but whose sodium deficit was similar to the one measured in other athletes that did not develop hyponatremia [28,54]. In these subjects, the predicted serum sodium concentration post-race exceeded the measured concentrations, indicating that osmotically active sodium was lost possibly due to the inactivation of osmotically active sodium. Data showed that those who osmotically activated sodium during recovery finished the race with higher serum sodium than did those who osmotically inactivated sodium during recovery. However, the hypothesis of these stores was not supported in an animal study [77]. Τhe exact underlying mechanisms for the retention of excess fluid because of inadequate suppression of antidiuretic hormone secretion, and inactivation of circulating osmotically active sodium or failure to mobilize osmotically inactive sodium from internal stores, are not completely understood [31].

Water remaining in the gut has been identified as a factor affecting the risk for EAH. During high water intake, water remains in the intestinal tract which leads to sodium being transferred from the blood into the gut [32,66,78]. Considering these effects, studies show that water alone is absorbed into the bloodstream slower than beverages with carbohydrates [79]. Water absorption is influenced by osmolality and solute absorption, while different types of carbohydrates in a sports drink with carbohydrates and electrolytes can help absorption and decrease of osmolality in the intestinal track [80]. This effect can explain the lower rate of water or hypotonic fluid absorption into the bloodstream, which can lead to sodium being absorbed into the intestinal lumen. During exercise optimal water absorption from the gut is important, hence sports drinks are formulated in a way that both electrolytes, carbohydrates for fuel and water are absorbed fast [81].

## 4. The Clinical Presentation of Exercise-Associated Hyponatremia

Hyponatremia is defined biochemically when a plasma sodium concentration of <135 mmol/L is measured [53,82]. A mild and slow onset of EAH does not cause long-term symptoms, and the condition of EAH may also be asymptomatic [45,83] because in many cases EAH remains clinically unremarkable [53,83] and performance is not impaired [84]. Performance is limited in only rare cases of EAH [38]. A manifest clinical symptomatology of EAH is generally to be expected with a plasma sodium concentration of <120 mmol/L in plasma [85]. Chronic EAH up to a concentration of about 115 mmol/L will remain asymptomatic [86]. In endurance athletes, a severe clinical manifestation with a plasma sodium concentration of <130 mmol/L is present in only ~1.5% of all cases [53]. A rapid decrease in plasma osmolality leads to a water influx across the blood-brain barrier and results in a cerebral edema. During prolonged endurance performance (e.g., longer than 10 h), symptoms of hyponatremia may be different after five to six hours, since hyponatremia can develop fast or slow [12,13,87]. Table 1 summarizes the possible symptoms of hyponatremia: malaise, mild headache, vomiting, and fatigue appear in the early stages of EAH [9,12,53]. These symptoms are very non-specific and can also occur with other problems such as fatigue, indigestion, dehydration, or overheating. The clinical appearance of hyponatremia may look like heat stroke, hypoglycemia, stress-related collapse, muscle cramps [88], or even altitude sickness [89]. The plasma sodium concentration should then have fallen to values <125 mmol/L. Symptoms at a plasma sodium concentration >125 mmol/L are rare. In some cases, vomiting is the only clinical sign that distinguishes hyponatremia from exercise-induced collapse [90].

Hyponatremia symptoms are non-specific, vary between patients, and may be confused for other conditions. Symptomatic EAH can occur at sodium concentrations around 130 mmol/L, as it has been shown in various case reports [11]. The symptoms depend not only on the serum sodium concentration, but also the decrease rate [13]. Symptomatic EAH can occur if the rate of fall approaches 7% to 10% within 24 h [91]. Thus, more severe degrees of hyponatremia (typically, 125 mmol/L) as well as more modest serum sodium values (in the range of 125–130 mmol/L), that develop over a short period of time, can both be associated with signs and symptoms [92]. 

As EAH increases with a decrease in plasma sodium concentration, cerebral and pulmonary edema will develop [93]. With the formation of cerebral edema at a plasma sodium concentration of <120 mmol/L, symptoms such as headache, confusion, agitation, coordination problems, and unconsciousness can occur [9,12]. Several cases of long-distance runners with symptomatic EAH are known that include altered behavior, seizures and edema [24,94,95,96,97]. When the plasma sodium concentration decreases to <110–115 mmol/L, symptoms such as muscle twitching, disorientation, coma, and epileptic seizures can occur. However, case studies have shown that such symptoms can occur at concentrations around 116–130 mmol/L [11].

If it decreases further, severe respiratory failure, including respiratory arrest can develop [12,85,98]. Unconsciousness can be due to increased intracranial pressure or pulmonary edema. In extreme cases, death will occur when the symptoms are not recognized and when no adequate treatment is established [94,99]. Excessive hydration during exercise also has a negative impact on the volume of the feet. Recent studies in ultra-marathoners showed an association between fluid intake and foot swelling [100,101]. Fluid overload during a competition is just one of several possible causes of dying from hyponatremia. 

## 5. The Prevalence of Exercise-Related Hyponatremia

Until now, it has been assumed that dehydration is inevitable during prolonged physical exertion [48]. Therefore, it has been recommended that as much fluid as possible should be consumed during exercise to prevent serious problems such as heat stroke or kidney failure. Until a few decades ago, the possibility of water poisoning with potentially fatal outcomes was considered as non-realistic. In recent literature, EAH has been considered potentially life-threatening [4,5,6,102], as some deaths of women marathon runners due to EAH have been reported [1,61]. The most common cases of EAH were subsequently detected in Ironman triathlons and ultra-marathons [29,34,47,65,66,103]. Up to 30% of the athletes who completed such competitions had laboratory-confirmed hyponatremia [34,53,104,105,106]. EAH was then considered to be a widespread and sometimes serious disorder in ultra-long exposures [53].

### Risk Factors for Exercise-Associated Hyponatremia

There appear to be specific event and/or athlete risk situations where EAH occurs more frequently (Table 2) [9,12,13,55,87,90,91]. EAH with a plasma sodium concentration of <130 mmol/L occurs especially in long to very long endurance performances [44] and slow and weak participants [15,53,90], as well as women [53,107]. EAH, however, is not limited to long-distance runners or triathletes. Descriptions of hyponatremia in soldiers [108,109], yoga [8,110], bowling [111], tennis [112], American football [13,113], rugby [114], river rafting [96], hiking [115,116], trekking [117], spinning [118], and rowing [119] do exist. Surprisingly, symptomatic EAH can also occur in children [120] and adolescents who are exercising with moderate intensities [98]. In a recent case report of a 15-year-old runner, a mild and asymptomatic EAH was present after a self-paced marathon [121].

Low sodium intake alone does not seem to affect the risk of hyponatremia as much as fluid over-drinking [83]. However, low dietary sodium intake for 10 days before a race can cause reductions in plasma sodium concentration, which in conjunction with fluid losses and high fluid consumption may lead to adverse effects during the race [122]. Those who fail to lose 0.75 kg are seven times more likely to be hyponatraemic than those who lose >0.75 kg [123].

## 6. The Prevalence of Exercise-Associated Hyponatremia by Sports Discipline

The prevalence of hyponatremia has been reported in various sports disciplines such as swimming [124,125,126], road cycling [69,126,127], mountain biking [126,128,129], running [62,126,130,131,132], triathlons [32,47,127], and rowing [119]. 

### 6.1. Swimming

When looking at different sports disciplines (e.g., swimming, cycling, mountain biking, running), swimming seems to present the highest prevalence of EAH [125,126,133]. Open-water swimming over longer distances seems to have risks, especially for women. At the 26.4-km long Marathon Swimming in Lake Zurich, Switzerland, four out of 11 women (36%) and only two out of 25 men (8%) showed EAH [125]. In a case report of a female open-water ultra-distance swimmer, exercise-associated hyponatremic encephalopathy presenting as altered conscious state and seizures was reported after she had completed a 20-km open ocean swim, with a serum sodium concentration of 119 mmol/L about one hour after her seizure [133]. 

### 6.2. Cycling

For road and mountain bike cyclists, a rather low prevalence of EAH has generally been found [69,126,129]. In a bicycle road race over 210 km and 250 km, 4 out of 90 subjects (4.5%) developed EAH [69]. In road cyclists competing in 109-km cycling race, 12% developed EAH [41]. In a study on 50 mountain bikers competing in a 24-h race, six athletes with mild EAH (3%) were reported [134]. In the Swiss Bike Masters, a mountain bike race over 120 km and covering an altitude difference of around 5000 m, no case of EAH was detected in 37 athletes [128]. Likewise, no case of EAH could be detected in the Jeantex Bike Transalp over eight stages covering 665.40 km and 21,691 vertical m [129]. For road cyclists [126,127] and mountain bikers [126,128,129], the low prevalence of EAH may be due to their specific nutrition supplies. In contrast to swimmers and runners, cyclists have their drink bottles on their bicycles and can thus cater to their individual needs. Although there is a relative lack of studies concerning EAH in sports apart from triathlon and marathons, current reports do show low incidence of EAH in cyclists, possibly due to low fluid intakes [128,135]. Considering endurance swimming, there have been some case reports of hyponatremia, especially in women [125,133]. It is also astonishing to realize that faster mountain bikers drink more than slower bikers and the prevalence of EAH is not increased. In very rare circumstances, EAH can occur in cyclists, as seen in a published case report of an experienced male cyclist showing symptoms of EAH as encephalopathy [25]. There may also be cases of EAH in mountain biking. One case report described a 44-year-old man with a plasma sodium concentration of 116 mmol/L after a 100-mile mountain bike race in Leadville, Colorado, USA [136].

### 6.3. Running

It appears that the prevalence of EAH increases with increasing duration of an exercise [9,93,130]. In a half -marathon (21.1 km) no case of hyponatremia was detected in 130 runners [137]. For marathons (42.2 km), the mean prevalence for EAH is around 8% (Table 3). The prevalence of EAH in marathons held in the USA is especially high. When we look at the prevalence of ultra-marathon races (Table 4), there is a difference between the runs shorter than 100 km, over 100 km, and over 100 miles. For runs shorter than a 100-km ultra-marathon, the prevalence of EAH is on average <1%, thus significantly lower than at a classical marathon event. In 100 km runs, the prevalence is <3%, thus also significantly lower than at marathons. In the 100 mile runs, the mean prevalence of EAH is well over 20%, thus also above the mean value for marathons. Obviously, the prevalence of EAH in a marathon held in the USA (e.g., Boston, Houston) is high and the 100-mile runs were exclusively held in the USA. According to personal experience of an ultra-marathoner competing in the USA, the Swiss ultra-runner Christian Marti was weighed at the start and in the middle of the track during the 100-mile race in Vermont (USA). When body weight would have fallen below a certain value, he would have had to refill the weight loss with the ingestion of fluids on the spot (personal message Christian Marti).

The duration of exercise is critically important for potentially life-threatening EAH [102]. As a rule, exposure to the development of EAH lasts more than five hours, and athletes consume a lot of fluids during exercise [45,90] especially during the first few hours [139]. EAH is less likely, yet it can be expected for exposures of less than four hours [1,140], but is very likely to occur in endurance performances enduring longer than eight hours [34]. However, asymptomatic hyponatremia was observed also in 33% of UK rugby players following rugby match of 80 min [114]. 

In a marathon, about 5% [82] to 15% of the runners [61] may experience EAH. Depending on the data collection, however, the prevalence of EAH in a marathon may also be <5% [59,90,138] but may also reach over 20% [123]. In some cases, EAH may be dramatic after a marathon [89,141,142]. Due to the nature of the studies and case reports, the precise prevalence of EAH cannot be estimated. One case report describes a 23-year-old man completing a marathon in just over four hours. He complained of weakness and dizziness at the finish and vomited after the race. EAH was detected and cerebral edema was found in cCT (cranial computed tomography). The patient needed intensive care treatment for six days [141]. Another study reported seven marathon runners who developed EAH after running, with hyponatremic encephalopathy and non-cardiac pulmonary edema in two cases [89]. Two decades ago, in the ‘Suzuki Rock ‘N’ Roll Marathon’, 26 runners were hospitalized with EAH, including 15 cases of severe hyponatremia with plasma sodium concentration < 125 mmol/L). In the severe cases, three patients developed seizures and had to be intubated [142].

Results are different for runs that are longer than a classical marathon distance. During an ultra-marathon, weight loss due to fluid loss is expected [64] and a large number of runners experience dehydration [143,144]. In a study by Hoffman et al. 35.6% of marathon runners were dehydrated [57]. When an ultra-marathon is carried out as a multi-stage run, the dehydration from stage to stage increases even more [62].

Ultra-marathoners have to consume large amounts of fluids during a race to avoid dehydration in terms of weight loss [155]. It has been reported that the greatest weight loss was recorded in the first few hours of an ultra-marathon [139]. However, in an ultramarathon, there is no need to drink large amounts of fluid [156]. Even when ultra-runners lose more than 3% of their body weight during a run, there is no reason to over-drink to prevent overheating [157]. In general, ultra-runners do not seem to consume excessive fluid [128] and ultra-marathoners should not experience fluid overload while competing [158]. For example, faster runners drank more liquid in a 100-km ultra-marathon than slower runners, and faster runners lost more weight than slower runners [158]. However, weight loss was greater with less fluid intake [156]. Due to the fact that faster runners drank less and lost more weight, the weight loss can be ergogenic (i.e., performance enhancing) and the runners achieve a faster race time [156]. In a 100-km ultra-marathon, it was even shown that runners with a greater weight loss were faster [127]. Even in a 100-mile ultra-marathon, major weight losses were more likely to boost performance [159]. On the other hand, weight loss is not due to fluid loss only, as it is also due to macro-nutrient use for energy [64,160].

Ultra-runners also seem to be able to self-regulate their plasma sodium concentration during a run [55]. In an ultra-marathon, the prevalence of EAH was much higher during the race than after the race [55]. It has also been reported that runners with the highest fluid intake had the lowest hemodilution with no changes in plasma sodium and potassium [161]. EAH often occurs during an ultra-marathon, but, in contrast to other sports disciplines, only in very few cases do major medical problems occur. Thus, a case of a 57-year-old man who developed EAH during a 100-mile ultra-marathon is described. Hyponatremia resulted in rapid neurological deterioration and cardiovascular instability [24].

### 6.4. Triathlon

Long-distance triathlons are about the same as an ultra-marathon. In an Ironman-triathlon (i.e., 3.8 km swimming, 180 km cycling, and 42.2 km running), EAH can be detected in about 20% of the athletes [52]. In the ‘Ironman Hawaii’, EAH is the most important electrolyte disorder [34]. About 9% of athletes who collapsed during an Ironman-triathlon had hyponatremia [44]. However, only about 30% of athletes with proven hyponatremia in the laboratory needed medical treatment [53]. There are also longer triathlon competitions than the Ironman-triathlon. In a triathlon covering three times the Ironman distance (i.e., 11.4 km swimming, 540 km cycling, and 126.6 km running), a higher prevalence of EAH of 26% was detected than is reported for Ironman triathletes [127].

### 6.5. Multi-Stage Events

Several studies have measured plasma sodium concentration during multi-stage races [62,129,131]. It appeared that hyponatremia can be corrected quite well during multi-stage races until the arrival of the finish [129,131]. However, when a multi-stage running race takes place in a desert, the prevalence of EAH increases from stage to stage [62]. An intense training camp can also be considered as a multi-stage event. For example, rowers in a four-week training camp showed that on the 18th day of the camp, the training volume and prevalence of EAH was highest at 43% [119].

## 7. Prevalence of Exercise-Associated Hyponatremia Regarding Ambient Temperatures

EAH is a relatively frequent, detectable electrolyte disorder in ultra-distance performances [56,62,130,149,153,162,163], whereby high ambient temperatures must be given great importance [57,154,162,164,165,166]. 

### 7.1. The Aspect of Heat

For endurance exercise in high temperatures, EAH can occur even after a relatively short period of stress [116,164,167]. Three soldiers who had to march in great heat showed symptoms of EAH in less than three hours [167]. There seems to be an increased risk for EAH in races held under thermally stressing conditions such as great heat [57,154,162] and great cold [168] especially in the USA. In a study in a 100-mile ultra-marathon held in California, the prevalence of EAH was clearly correlated with ambient temperature [57]. The prevalence of EAH was 42% during a 225-km ultra-marathon over five stages with ambient temperatures as high as 40 °C [162]. In the Rio de Lago 100-Mile Endurance Run of 2008, held in Granite Bay, California, the prevalence of EAH was 51.2%, accounting for about half of all finishers [154]. However, EAH in 100-mile runners can be as low as 30% [56]. 

### 7.2. The Aspect of Humidity

Humidity, however, could also play a role. During a running event in the tropics with seven different routes (i.e., 10, 21, 25, 42, 50, 84, and 100 km), EAH was detected in eight cases (17%): four runners in the marathon, two runners in the double marathon (84 km), and two runners in the 100 km run [166]. 

### 7.3. Exercise-Associated Hyponatremia in the Cold

Great cold also seems to be a risk contributing to the development of EAH [14,169]. Stuempfle et al. investigated EAH in races held under very cold conditions [63,168]. In a 100-mile ultra-marathon held in Alaska, 44% of runners presented with EAH [168]. Runners developing EAH consumed more fluid and less sodium than those without EAH [168]. However, in another Alaskan race, no cases of EAH could be detected. In 21 athletes (i.e., 11 runners, six cyclists, and three cross-country skiers) in a 100-mile competition of the three disciplines, plasma sodium concentration decreased after the competition, but no case of EAH occurred [63]. Basically, endurance athletes drink very little when it is very cold. 

In open-water swimming in moderate temperatures, the prevalence of EAH seems to be rather high. At the 26.4-km long Marathon Swimming in Lake Zurich, Switzerland, the prevalence of EAH was about four times higher in women compared to men [125].

### 7.4. Exercise-Associated Hyponatremia in Moderate Ambient Temperatures

In ultra-marathons held in temperate climates, EAH is relatively uncommon [126,127,131,132,150,158,170,171]. For example, ultra-marathons held in Switzerland present a relatively low prevalence of EAH [126,127,131,132,150,158,170]. In the Swiss Jura Marathon, a mountain ultra-marathon of 350 km over 7 stages held in the Swiss Jura from Geneva to Basel and taking place in medium to low temperatures, the prevalence of EAH was 8% [131]. No case of EAH could even be detected in the 100 km Lauf Biel [132,158] or at the 24-Stunden-Lauf Basel [170], both held in medium to low temperatures. Even for cycling in temperate climates, the prevalence of EAH is very low [126]. In a road race in Switzerland over 720 km [135], as well as a mountain bike race over 120 km and an altitude difference of around 5000 m [128], no case of EAH have been detected.

## 8. Female Sex as Risk Factor for Exercise-Associated Hyponatremia

Due to the lack of high quality observational studies and randomized trials concerning the risks and incidence of EAH in men and women, as well as due to the relatively low prevalence of EAH, a series of case reports are currently evaluated. It is important to know that the first documented case of EAH in 1981 affected a woman in the Comrades Marathon [17]. It seems that weight-associated EAH is more common in women than in men [9,93,107,125,167] likely due to their different body mass. A possible cause of the increased prevalence of EAH in women could be the higher fluid intake in women as opposed to men [124], since hydration guidelines are common for both men and women, despite their different physiology. Under laboratory conditions, it has been shown that women drink more water than men and therefore develop EAH [107]. In addition, women have higher water retention during exercise than men [172]. However, women tended to drink less than men in the Houston Marathon, and had a prevalence of 22% for EAH [123]. In a 24-h run held in extreme cold, women drank no more than men [173]. The main role for women is most probably the lower body weight compared to men. A study performed in the Boston Marathon suggested that the apparent sex difference disappeared when data were adjusted for body mass index and racing times [61].

Instead of sex, lower BMI and racing times may be the reason why women seem to be at a higher risk of EAH than men. This was evident in a study in 488 Boston marathoners [61]. On the multivariate analysis, hyponatremia was associated with weight gain (odds ratio, 4.2; 95 percent confidence interval, 2.2 to 8.2), a racing time of >4:00 h (odds ratio for the comparison with a time of <3:30 h, 7.4; 95 percent confidence interval, 2.9 to 23.1), and body-mass-index extremes, but not sex. Women are slower than men, hence they also have longer racing times [174].

EAH, however, does not always have to be more common in women than in men. In the 100 km Lauf Biel held in Switzerland, only one out of 19 women (5%) and three out of 24 men (11%) developed EAH [150]. And in another study of the 100 km Lauf Biel, no case of EAH could be detected in the 11 women who were observed [146]. Often, EAH in women is described in somewhat dramatic case reports [96,110,118,120,167,175]. One case report described a 49-year-old woman with anorexia, who developed excessive rhabdomyolysis and hyponatraemia through excessive and compulsive exercise, although she did not consume excessive fluids [176]. In another case, a female open-water swimmer was mentioned, who was admitted to the hospital after a 20 km swim with disturbances of consciousness and seizures. There, evidence of EAH with encephalopathy was achieved at a plasma sodium concentration of 119 mmol/L. After intensive care treatment, she was able to be discharged healthy and without neurological impairment [133].

Several cases of female triathletes with EAH were described [175,177,178]. A 42-year-old woman was hospitalized after an Ironman triathlon with headache, nausea, and confusion. Over time, she developed seizures, cerebral edema could be detected in cCT, and plasma sodium concentration was 123 mmol/L. Subsequently, there was a dramatic worsening with a GCS of 3 and intensive care treatment with intubation, forced diuresis, and measurement of intracranial pressure was established. After 16 h, plasma sodium concentration returned to normal levels, after two days the patient was extubated, after one month memory problems were still present, and only six months later was she able to return to work [178]. Consider the case of a 19-year-old triathlete who tackled her first triathlon of 400 m of swimming, 17 km of cycling and 5 km of running. She drank 3 l the night before the competition, 0.5 l during the race, and after the competition another 0.7 l. As a result, she developed symptomatic EAH leading to hospitalization [177]. In another case report, a 45-year-old female Ironman triathlete presented with somnolence and convulsions after finishing her first race. Besides the neurological symptoms, she had a swollen face and swollen ankles. The laboratory tests showed severe hyponatremia with plasma sodium concentration of 111 mmol/L. Radiological examination revealed pulmonary as well as cerebral edema [175].

Two cases of marathoners with symptomatic EAH have been reported [179,180]. In one case, a 52-year-old marathon runner was described who developed EAH after the run and then non-cardiac pulmonary edema [179]. In another case, a 41-year-old marathon runner developed EAH and brain edema during the marathon [180]. In addition to classic endurance exercise, other physical stresses can trigger hyponatremia in women. One case described a 34-year-old woman who developed intense dyspnea, muscle spasms, and nausea after intense yoga in a temperature above 40 °C, due to heavy sweating. In the hospital, a hyponatraemia of 120 mmol/L was detected, seizures occurred, and the patient had to be intubated [8]. In rare cases, EAH is fatal for females. One case report described a 9-year-old girl who was forced to repeat sprints by her grandmother for stealing candy from a schoolmate. After three hours of running, the girl collapsed, vomited, and experienced epileptic seizures. In the hospital, a hyponatraemia with a plasma sodium concentration of 117 mmol/L was detected and despite treatment, the child died. The autopsy revealed massive brain and pulmonary edema [120]. Several cases of hyponatraemia have also been reported in the Grand Canyon in recent years [181]. A case report described a woman collapsing in the Grand Canyon National Park after a five-hour hike tour during which she consumed large amounts of fluid. She was admitted to the hospital unconscious, where hyponatraemia and hyponatremic encephalopathy was detected. Within 24 h she died of cerebral edema [182]. In another case report from Grand Canyon National Park, three women who were rafting on the river were admitted to hospital for fatigue, vomiting, and disorientation, where symptomatic hyponatremia required intensive care [96]. Due to these cases of hyponatraemia outside of organized sports events, it is now increasingly being pointed out that symptomatic EAH can also occur in activities other than athletic competitions [183,184,185,186].

However, there are also less dramatic situations of EAH in women [187]. Women can perform well despite developing EAH. In one case description, three women were described as winning 24-h races in different conditions and disciplines ahead of all men. In all three cases, mild and asymptomatic EAH was found [187]. A possible explanation for the increased prevalence of EAH in women could be Varon–Ayus syndrome [188]. This syndrome includes severe hyponatraemia and lung and cerebral edema, and was first described by McKechnie and colleagues [189], and thereafter, by Varon and Ayus in marathon runners [85,190]. In principle, the situation can be fatal if it is not recognized [190], but can be easily resolved by infusing a hypertonic NaCl solution [85,190]. Varon–Ayus syndrome distinguishes the Ayus–Arieff syndrome from hyponatremic encephalopathy and non-cardiac pulmonary edema [191]. In contrast to Varon–Ayus syndrome, Ayus–Arieff syndrome usually occurs postoperatively. Women and men are at the same risk of developing hyponatremia and hyponatremic encephalopathy after surgery. However, considering hyponatremic encephalopathy, menstruant women are at 25 times higher risk of death or permanent brain damage, compared with either men or postmenopausal women [91]. This effect was due to irrigant absorption hyponatremia after prostate surgery in men.

## 9. The Prevalence of Exercise-Associated Hyponatremia (EAH) by Region Where the Race is Held

The country where a marathon [123] or an ultramarathon takes place seems to be of utmost importance [56,126,130,146,150,170]. For example, the prevalence of EAH in events held in the USA is much higher [56,123,130,153,167] than in events held in Europe [126,150,170]. While the prevalence of EAH in the Western States Endurance Run held in California was at around 16% [153] to 30% [56], the prevalence of EAH did not exceed 11% in ultra-marathons held in Switzerland [126,131,132,149,150,170]. In endurance events held in the Czech Republic, the prevalence of EAH was very low [134,192,193,194]. In the 113 recreational athletes, the prevalence of EAH was 11.5%, with all athletes presenting only mild hyponatremia [134]. In another group of the 26 ultra-runners running seven marathons in seven consecutive days, the prevalence of EAH was 3.8% [194]. During a 24-h ultra-marathon held in winter, hydration status was maintained [173], with one athlete presenting mild EAH (4.8%) [187].

In other regions of the world, such as Asia, hyponatremia is rare during sports, and the prevalence of EAH in an 84-km ultra-marathon was only 3% [195]. The risk of developing EAH seems to be very low in Australia and New Zealand. No case of EAH has been demonstrated at the Six Foot Track Mountain Ultramarathon held in New South Wales [145]. At the Cradle Mountain Run in Tasmania, only 2% of runners developed EAH [147]. At the Kepler Challenge, a 60-km mountain ultra-marathon held in New Zealand, EAH was detected in five of 123 runners [196]. In South Africa, too, the risk of burden-associated hyponatremia seems to be very low. No case of EAH could be detected at the Two Oceans ultra-marathon held in Cape Town [146]. When looking at other disciplines, no cyclist has developed EAH in a 720 km road race held in Switzerland [127].

A very likely explanation for the higher prevalence of EAH in ultra-marathons held in the USA compared to ultra-marathons held in Switzerland, Europe, might be the ambient temperatures. When the Western States Endurance Run is held, the temperatures vary between 59 °F (15 °C) and 89 °F (32 °C) [197]. In contrast, the average temperature in the 100 km Lauf Biel held in Switzerland are about 10 °C lower [198]. It is very likely that runners drink more when the temperatures are higher and therefore the risk for fluid overload and EAH is higher.

## 10. Prevention of Exercise-Associated Hyponatremia

Prevention of EAH is of critical importance and requires organized educational programs with information disseminated to coaches, athletes, and event staff regarding healthy hydration practices, sodium supplementation, and recognition and treatment of EAH [1]. The best methods to prevent from fluid overload are drinking according to thirst, reducing the availability of fluids along the routes of exercise and monitoring weight changes during exercise [1]. Despite being a well-known condition by now, recent deaths due to EAH highlight the importance of guidance in order to prevent EAH [3]. Evidence-based advice and strategies that target the widespread misinformation should be available to amateur and professional athletes [199]. Educational strategies for optimal hydration practices should be encouraged [200]. 

The Statement of the 3rd International Exercise-Associated Hyponatremia Consensus Development Conference, Carlsbad, California states strategies to prevent EAH [13]. Prevention of over-hydration/over-drinking fluids is a priority, while mild dehydration at 3% of body weight is deemed tolerable, without reducing performance in temperatures between −10 to 20 °C. Drinking when thirsty can be an effective and safe strategy for optimal hydration, as opposed to the potentially dangerous advice “drink as much as possible”. An individualized plan according to the needs of the athlete, based on weight changes during training, may be an effective strategy, although considerations should be made in extreme environments [13]. Another strategy that has been implemented with positive results is the lower availability of fluids, however, a higher availability may be needed in extreme conditions [140,201].

A further aspect in this regard is the type of rehydration and the importance of taking drinks able to reconstitute the correct electrolytic level in these conditions. While sodium ingestion during a race may attenuate the fall in blood sodium concentrations, it cannot prevent EAH in the setting of excessive fluid intake [1]. It is the amount of fluid ingested rather than the amount of sodium ingested during exercise that drives the final blood sodium concentrations. Sodium-containing sports drinks, which are hypotonic, will not prevent EAH in athletes who overdrink during exercise [1].

## 11. Conclusions

The prevalence of EAH depends on the duration of an endurance performance (i.e., low in marathon running, but high to very high in ultra-marathon running), the sports discipline (i.e., very rare in cycling, increased in running and triathlon, and occasionally very high in swimming), sex (i.e., increased in women with several reported deaths), the ambient temperature (i.e., very high in heat), and the country where the competition takes place (i.e., very common in the USA, very little in Europe, practically never in Africa, Asia, and Oceania). A possible explanation for the increased prevalence of EAH in women could be Varon–Ayus syndrome with severe hyponatraemia and lung and cerebral edema, which was first described in marathon runners. Also, in women, the lower body mass might lead to an increased risk. Regarding the race location, races in Europe seemed to be held under rather moderate conditions, whereas races held in the USA were often performed under thermally stressing conditions.

## Figures and Tables

**Table 1 medicina-55-00537-t001:** Symptoms of hyponatremia.

**Fast Development (Brain Edema)**
Headache
Nausea
Vomiting
Dizziness
Weakness
Adynamia
Fatigue
Tremor
Epileptic seizures
Muscle cramps
Increase in body weight
Swelling of hands and feet
Somnolence
Coma
**Slow to chronic development**
Fatigue
Disorientation
Lethargy
Confusion
Inappetence
Change of personality
Gait disturbance
Attention deficit disorder

**Table 2 medicina-55-00537-t002:** Risk factors for exercise-associated hyponatremia.

**Athlete**
Female sex (especially menstruating)
Short stature
Heavy and excessive drinking (seen as weight gain during exercise)
Low body weight
Low BMI
Weight gain during exercise
Slow running pace
Low competition experience
Intake of NSAIDs
**Event**
Duration of four hours and longer
High availability of fluids
Extreme heat
Extreme cold

**Table 3 medicina-55-00537-t003:** Prevalence of exercise-associated hyponatremia (EAH) in marathon races.

Race	Prevalence of EAH
Houston Marathon 2000 [90]	<1%
Zürich Marathon [59]	3% (five of 167 subjects)
Boston Marathon 2001–2008 [138]	4.8% (in 1319 collapsed runners)
Marathon [82]	5.6%
Boston Marathon [61]	13%
Houston Marathon 2000–2004 [123]	22% (21 of 96 subjects)

**Table 4 medicina-55-00537-t004:** Prevalence of exercise-associated hyponatremia (EAH) in ultra-marathons.

Ultramarathon	Distance	Prevalence
**Below 100 km**		
Six Foot Track Marathon, New South Wales, Australia [145]	45 km trail	0% in 9 subjects out of 775 starters
Two Oceans 56-km ultra-marathon, Cape Town, South Africa [146]	56 km	0%
Cradle Mountain Run, Tasmania, Australia [147]	85 km	2% of 41 subjects
**100 km**		
100 km Lauf Biel, Switzerland [132]	100 km	0% of 50 men
100 km Lauf Biel, Switzerland [148]	100 km	0% of 11 women
100 km Lauf Biel, Switzerland [149]	100 km	4.8% of 145 subjects
100 km Lauf Biel, Switzerland [150]	100 km	5% in 19 women and 11 % in 24 men
100 km Lauf Biel, Switzerland [126]	100 km	5% of 95 men
**100 miles**		
Western States Endurance Run, California, USA [151]	100 miles	5.8% of 207 subjects
Western States Endurance Run, California, USA [152]	100 miles	6% of 207 subjects
Western States Endurance Run, California, USA [83]	100 miles	6.6% of 157 subjects
Western States Endurance Run, California, USA [57]	100 miles	15.1% of 669 subjects
Western States Endurance Run, California, USA [153]	100 miles	16% of 373 subjects
The Great North Walk 100s, New South Wales, Australia [55]	100 miles	26.7 % in 4 of 15 subjects
Western States Endurance Run, California, USA [56]	100 miles	30% of 47 subjects
Western States Endurance Run, California, USA [130]	100 miles	30% of 47 subjects
Rio Del Lago 100-Mile Endurance Run Granite Bay, California, USA [154]	100 miles	51.2%

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
