# Peer review of "Exercise-Associated Hyponatremia in Endurance and Ultra-Endurance Performance–Aspects of Sex, Race Location, Ambient Temperature, Sports Discipline, and Length of Performance: A Narrative Review"

_medicina, 2019, doi:10.3390/medicina55090537_

Round 1

Reviewer 1 Report

My advice is to describe that this is a narrative/literature review and also include an objective/aim of the research in the Abstract

From my point of view, the information regarding search strategy could be described in a new section (methods), in which the authors could also mention how many studies were found and how many were discarded. Some inclusion/exclusion criteria would be useful too.

The authors state that   When plasma or serum concentration of sodium is <135 mmol/l, hyponatremia is diagnosed and then they repeat this information later on. I think that once is enough.

Check consistency when writing, sometimes the authors use Sodium and in other occasions they use Na+

The title of table 1 and table 2 should not include references

Tables 3 and 4, place references in the first column, just after the name of the race

I would first introduce the prevalence of hyponatremia by discipline and then I will focus on marathon and long distance running competitions.

May be the authors could think about introducing one section (i.e. Risk factors for hyponatremia, Factors related to hyponatremia…) and several subsections such as type of discipline, long distance run, heat, sex… (by the way the authors could think about rephrasing “the aspect of sex”).

The influence of the country in which the competition takes place is an interesting factor.

The authors could opt for providing some explanations regarding why in some countries the prevalence is higher than in others.

I strongly advise to introduce a section about preventive strategies. What can be done in order to prevent the onset of hyponatremia. This is an important issue.

Author Response

Comments and Suggestions for Authors

My advice is to describe that this is a narrative/literature review and also include an objective/aim of the research in the Abstract

Answer: We agree with the expert reviewer and add the aspect of narrative review in the title. We further added in the abstract ‘The objective of this narrative review is to present new findings about the aspects of sex, race location, sports discipline and length of performance’

From my point of view, the information regarding search strategy could be described in a new section (methods), in which the authors could also mention how many studies were found and how many were discarded. Some inclusion/exclusion criteria would be useful too.

Answer: We agree with the expert reviewer and added a separate section as requested. It reads now ‘We considered the data bases PUBMED and SCOPUS for literature search by using the terms ‘exercise-associated hyponatremia’ (237) with the combinations ‘sex’ (16), ‘origin’ (1), ‘race’ (81), ‘discipline’ (3), ‘swimming’ (17), ‘cycling’ (14), ‘running’ (100), ‘mountain bike’ (7), ‘endurance’ (114), and ‘ultra-endurance’ (33). Both case reports and research articles were considered. Studies regarding physiological and pathophysiological aspects were not considered’.

The authors state that   When plasma or serum concentration of sodium is <135 mmol/l, hyponatremia is diagnosed and then they repeat this information later on. I think that once is enough.

Answer: We agree with the expert reviewer and changed to ‘EAH describes the occurrence of hyponatremia in individuals during prolonged exercise (i.e. usually longer than six hours) and is defined when plasma or serum concentration of sodium is <135 mmol/l’.

Check consistency when writing, sometimes the authors use Sodium and in other occasions they use Na+

Answer: We agree with the expert reviewer and changed to ‘sodium’ throughout the manuscript.

The title of table 1 and table 2 should not include references

Answer: We agree with the expert reviewer and changed as suggested

Tables 3 and 4, place references in the first column, just after the name of the race

Answer: We agree with the expert reviewer and changed as suggested

I would first introduce the prevalence of hyponatremia by discipline and then I will focus on marathon and long distance running competitions.

Answer: We agree with the expert reviewer and changed as suggested. We make now a section by discipline, including running of different lengths.

May be the authors could think about introducing one section (i.e. Risk factors for hyponatremia, Factors related to hyponatremia…) and several subsections such as type of discipline, long distance run, heat, sex… (by the way the authors could think about rephrasing “the aspect of sex”).

Answer: We agree with the expert reviewer and change the title to ‘Female sex as risk factor for exercise-associated hyponatremia’. We also have now a new section for the risk factors and an own section for the aspect of ambient temperature.

The influence of the country in which the competition takes place is an interesting factor.

Answer: We agree with the expert reviewer and added the aspect that the average temperatures are about 10 °C lower in Switzerland (100 km Lauf Biel) compared to USA (Western States Endurance Run).

The authors could opt for providing some explanations regarding why in some countries the prevalence is higher than in others.

Answer: We agree with the expert reviewer and added the aspect that the average temperatures are about 10 °C lower in Switzerland (100 km Lauf Biel) compared to USA (Western States Endurance Run).

I strongly advise to introduce a section about preventive strategies. What can be done in order to prevent the onset of hyponatremia? This is an important issue.

Answer: We agree with the expert reviewer and added a separate section on the prevention of EAH.

Reviewer 2 Report

The premise of this review paper is an interesting one, but the scientific rigour does not stand up sufficiently to scrutiny in its present form. 

There are numerous examples of disordered paragraphs were the logic of the arguments is not clear, where paragraphs appear to be made up of blanket conclusions from other studies with no obvious attempt to link them through discussion. These didactic statements allow no room for nuance or complexity within a clearly complex syndrome.  

There are some statements which are scientifically inaccurate yet which form the basis for a whole paragraph. Within paragraphs there are numerous statements which bear no relevance or links to the preceding or subsequent statements

There is little coherent planning of the sequence of arguments making it common to read numerous repetitions, including repetitions which contradict or undermine the positions stated initially. 

Towards the end of the paper the writing style changes to more of a commentary and discussion but even then it is still characterised by lists of statements without obvious links between them. 

It is recommended that this paper is extensively rewritten, paying careful attention to the logic and sequence of argument used. It is recommended that a more critical analysis and discussion of the evidence is undertaken rather than simply restating the evidence without analysis. 

Author Response

Comments and Suggestions for Authors

The premise of this review paper is an interesting one, but the scientific rigor does not stand up sufficiently to scrutiny in its present form. 

Answer: We agree with the expert reviewer and have improved the manuscript following the other three reviewers.

There are numerous examples of disordered paragraphs were the logic of the arguments is not clear, where paragraphs appear to be made up of blanket conclusions from other studies with no obvious attempt to link them through discussion. These didactic statements allow no room for nuance or complexity within a clearly complex syndrome.  

Answer: We agree with the expert reviewer and have improved the manuscript following the other three reviewers.

There are some statements which are scientifically inaccurate yet which form the basis for a whole paragraph. Within paragraphs there are numerous statements which bear no relevance or links to the preceding or subsequent statements

Answer: We agree with the expert reviewer and have improved the manuscript following the other three reviewers.

There is little coherent planning of the sequence of arguments making it common to read numerous repetitions, including repetitions which contradict or undermine the positions stated initially. 

Answer: We agree with the expert reviewer and have improved the manuscript following the other three reviewers.

Towards the end of the paper the writing style changes to more of a commentary and discussion but even then it is still characterized by lists of statements without obvious links between them. 

Answer: We agree with the expert reviewer and have improved the manuscript following the other three reviewers.

It is recommended that this paper is extensively rewritten, paying careful attention to the logic and sequence of argument used. It is recommended that a more critical analysis and discussion of the evidence is undertaken rather than simply restating the evidence without analysis. 

Answer: We agree with the expert reviewer and have improved the manuscript following the other three reviewers.

Reviewer 3 Report

The manuscript is interesting and highlights aspects, sometimes little considered, which are very important in the preparation and / or planning of an extreme / very extreme physical activity.
It is evident that the place, the climatic conditions but also the type of physical stress that is faced can condition the physiological response to the stress. Also the gender differences have to be taken in count planning these extreme activities.

A fundamental aspect that could perhaps be a little more highlighted is the type of rehydration and the importance of taking drinks able to reconstitute the correct electrolytic level in these conditions.

I suggest the authors to implement , where possible, the comments on hyponatremia in extreme cold conditions, and in swimming, activity where even for less important distances this dangerous physiological phenomenon is particularly present.

Author Response

Comments and Suggestions for Authors

The manuscript is interesting and highlights aspects, sometimes little considered, which are very important in the preparation and / or planning of an extreme / very extreme physical activity.

It is evident that the place, the climatic conditions but also the type of physical stress that is faced can condition the physiological response to the stress. Also the gender differences have to be taken in count planning these extreme activities.

Answer: We agree with the expert reviewer, no further changes are required for this point

A fundamental aspect that could perhaps be a little more highlighted is the type of rehydration and the importance of taking drinks able to reconstitute the correct electrolytic level in these conditions.

Answer: We agree with the expert reviewer and added this aspect in a new section ‘Prevention of exercise-associated hyponatremia’.

I suggest the authors to implement, where possible, the comments on hyponatremia in extreme cold conditions, and in swimming, activity where even for less important distances this dangerous physiological phenomenon is particularly present.

Answer: We agree with the expert reviewer and consider this aspect in a new section ‘Exercise-associated hyponatremia in the cold’.

Reviewer 4 Report

This is a condition is a common complication of especially endurance exercise due to a combination of over drinking beyond thirst and non-–osmotic arginine vasopressin release. 

It is important for the sports medicine community to understand the background provided by this manuscript in terms of providing information on the epidemiology of this condition, which can be life-threatening, including a fatality at a recent European Ironman triathlon championships.
This article therefore complements and updates the review on this condition in front years medicine for: 21.  DOI: 10.3389/F med.  2017.00021.

Ref 13 might be highlighted as the report of the 3rd EAH Concensus Conference with recommendations on prevention and emergent treatment.

Author Response

Comments and Suggestions for Authors

This is a condition is a common complication of especially endurance exercise due to a combination of over drinking beyond thirst and non-–osmotic arginine vasopressin release. 

Answer: We agree with the expert reviewer and added this aspect in the beginning of section 4.

It is important for the sports medicine community to understand the background provided by this manuscript in terms of providing information on the epidemiology of this condition, which can be life-threatening, including a fatality at a recent European Ironman triathlon championship.
Answer: We agree with the expert reviewer and cite this fatal case as a reference from a newspaper report

This article therefore complements and updates the review on this condition in front years medicine for: 21.  DOI: 10.3389/F med.  2017.00021.

Answer: We agree with the expert reviewer and use this article as a reference

Ref 13 might be highlighted as the report of the 3rd EAH Consensus Conference with recommendations on prevention and emergent treatment.

Answer: We agree with the expert reviewer and this paper is now used in a new section.